# Evaluation of Two-Month Antibody Levels after Heterologous ChAdOx1-S/BNT162b2 Vaccination Compared to Homologous ChAdOx1-S or BNT162b2 Vaccination

**DOI:** 10.3390/vaccines10040491

**Published:** 2022-03-23

**Authors:** Simone Barocci, Chiara Orlandi, Aurora Diotallevi, Gloria Buffi, Marcello Ceccarelli, Daniela Vandini, Eugenio Carlotti, Luca Galluzzi, Marco Bruno Luigi Rocchi, Mauro Magnani, Anna Casabianca

**Affiliations:** 1Department of Clinical Pathology, Azienda Sanitaria Unica Regionale Marche Area Vasta 1 (ASUR Marche AV1), 61029 Urbino, PU, Italy; simone.barocci@sanita.marche.it (S.B.); m.ceccarelli3@campus.uniurb.it (M.C.); daniela.vandini@sanita.marche.it (D.V.); 2Department of Biomolecular Sciences, University of Urbino Carlo Bo, 61029 Urbino, PU, Italy; chiara.orlandi@uniurb.it (C.O.); aurora.diotallevi@uniurb.it (A.D.); g.buffi@campus.uniurb.it (G.B.); luca.galluzzi@uniurb.it (L.G.); marco.rocchi@uniurb.it (M.B.L.R.); mauro.magnani@uniurb.it (M.M.); 3Department of Prevention, ASUR Marche AV1, 61029 Urbino, PU, Italy; eugenio.carlotti@sanita.marche.it

**Keywords:** SARS-CoV-2, COVID-19, heterologous vaccination, anti-S antibodies, IgG anti-S response

## Abstract

We evaluated the post-vaccination humoral response of three real-world cohorts. Vaccinated subjects primed with ChAdOx1-S and boosted with BNT162b2 mRNA vaccine were compared to homologous dosing (BNT162b2/BNT162b2 and ChAdOx1-S/ChAdOx1-S). Serum samples were collected two months after vaccination from a total of 1248 subjects. The results showed that the heterologous vaccine schedule induced a significantly higher humoral response followed by homologous BNT162b2/BNT162b2 and ChAdOx1-S/ChAdOx1-S vaccines (*p* < 0.0001). Moreover, analyzing factors (i.e., vaccine schedule, sex, age, BMI, smoking, diabetes, cardiovascular diseases, respiratory tract diseases, COVID-19 diagnosis, vaccine side effects) influencing the IgG anti-S response, we found that only the type of vaccine affected the antibody titer (*p* < 0.0001). Only mild vaccine reactions resolved within few days (40% of subjects) and no severe side effects for either homologous groups or the heterologous group were reported. Our data support the use of heterologous vaccination as an effective and safe alternative to increase humoral immunity against COVID-19.

## 1. Introduction

The COVID-19 pandemic has severely impacted the world in terms of health, society, and economy, and currently, vaccination is the most effective strategy to contrast SARS-CoV-2. Five vaccines have been authorized by the European Medicines Agency (EMA) and the Italian Medicines Agency (AIFA): Comirnaty (Pfizer-BioNTech), Spikevax (Moderna), Vaxzevria (AstraZeneca), COVID-19 Vaccine Janssen (Johnson&Johnson), and Nuvaxovid (Novavax). All except COVID-19 Vaccine Janssen require a two-dose vaccination schedule (primary vaccination), each at different time intervals. From December 2020, Italy started a vaccination campaign [1], and as of June 2021, the Italian vaccination program adopted age-specific restrictions for the use of Oxford-AstraZeneca ChAdOx1-S nCoV-19 (ChAd) due to uncertainties related to the risk of thrombosis with thrombocytopenia syndrome (TTS) [2].

In order to increase vaccination coverage, countries adapted their strategies, for example, regarding the interval between first, second and third dose or combination of adenoviral-vector and mRNA vaccines, based on the epidemiological situation and circulation of variants. The heterologous prime-boost strategy in which a different vaccine is given for the second dose in a recommended two-dose schedule is an alternative option to homologous vaccination. Evidence from studies on the heterologous vaccination strategy suggests that the combination of Vaxzevria and mRNA vaccines is safe and effective and induces a robust humoral response against SARS-CoV-2, allowing populations to be protected more quickly and improving the use of available vaccine supplies [3,4,5].

Here, we present results from a population-based serological survey in the northern area of the Marche region (Italy) to evaluate the antibody response induced by the heterologous ChAd and Pfizer-BioNTech BNT162b2 (BNT) and homologous prime-boost vaccine schedules (BNT/BNT and ChAd/ChAd).

## 2. Materials and Methods

### 2.1. Recruitment and Study Cohort Characteristics

The study participants (*n* = 1248) were recruited from professionally active healthcare workers (Azienda Sanitaria Unica Regionale—Area Vasta 1 ASUR Marche; *n* = 952) and university staff (University of Urbino Carlo Bo; *n* = 296) vaccinated against COVID-19 between December 2020 and June 2021 in Urbino (PU), Italy. A total of 184 (15%) subjects received the homologous ChAd prime-boost vaccination (group ChAd/ChAd), 985 (79%) received the homologous BNT prime-boost vaccination (group BNT/BNT) and 79 (6%) received the heterologous ChAd/BNT prime-boost vaccination (group ChAd/BNT) because on 14 June 2021, the Italian Ministry of Health issued an administrative act (Determina) on the use of vaccines Cormirnaty or Moderna in the mixed vaccination schedule (heterologous vaccination), in the subjects of age less than 60 years who had already received a first dose of the Vaxzevria vaccine. The cohort included 836 females (67%) and 412 males (33%). The mean (range) age of the study group was 51 (23–71). Thirteen individuals (1%) provided information about a positive COVID-19 diagnosis: 8 subjects between March and April 2020, so before the primary vaccination; 2 subjects from the BNT/BNT group between the second dose and the anti-S IgG test (two months after vaccination); 3 subjects did not provide information on the SARS-CoV-2 infection date. The characteristics of the three groups are resumed in Table 1. The serum of the vaccinated subjects was analyzed approximately two months after the second dose (mean ± SD, 62 ± 8 days).

### 2.2. Determination of Antibody Levels

Serum samples were stored at 2–10 °C for <4 days and processed at the Laboratory of Clinical Pathology (with certified quality management UNI EN ISO 9001:2015) of Urbino Hospital (ASUR Marche AV1). To determine the serological response of COVID-19 vaccines, anti-S IgG levels after vaccination were measured in serum samples with the CE IVD marked “LIAISON^®^ SARS-CoV-2 TrimericS IgG” chemiluminescence immunoassay (CLIA) (DiaSorin S.p.a., Saluggia VC, Italy). The assay was performed following the manufacturer’s instructions except for the dilution step required for specimens containing ≥2080 BAU/mL levels of anti-trimeric spike protein, which has been omitted. The manufacturer states a sensitivity of 98.7%, a specificity of 99.5% and assay range of quantification 4.81—2080 BAU/mL, with a cut-off for positivity at 33.8 BAU/mL. This assay allows detecting a wider antibody population produced by the immune response by determining IgG antibodies specific to the anti-trimeric spike protein against SARS-CoV-2, reducing the risk of false negative results. Moreover, as the manufacturer declares, the assay provides an indication of the presence of Neutralizing Antibodies (Nabs) against SARS-CoV-2, by showing a positive percent agreement of 100% (95%CI: 97.8–100.0%) and negative percent agreement of 96.9% (95%CI: 92.9–98.7%) in the Microneutralization Correlation test [6,7].

### 2.3. Additional Testing for Asymptomatic SARS-CoV-2 Infection

Healthcare workers were periodically monitored (every 15 days) with rapid antigen tests (LIAISON^®^ SARS-CoV-2 Ag, DiaSorin S.p.a., Saluggia VC, Italy) followed, if positive, by the real-time PCR (Simplexa COVID-19 Direct DiaSorin) on the same nasopharyngeal swab to check for RNA of SARS-CoV-2, following the manufacturer’s instructions.

For the university staff, additional testing was conducted on the remnant serum samples frozen on the day of blood collection. Nucleocapsid-specific IgM and IgG antibodies were tested to check for the possible contact of vaccinated subjects with the virus over the course of the study. The testing was performed with COVID-19 ELISA IgM and COVID-19 ELISA IgG kits (Diatheva srl, Cartoceto, PU, Italy), strictly to the manufacturer’s instructions. These assays allow for a qualitative detection of the antibodies. All subjects that tested positive for Nucleocapsid-specific IgM and/or IgG antibodies were checked for SARS-CoV-2 RNA in nasopharyngeal swabs by real-time PCR using the Diatheva COVID-19 PCR kit, following the manufacturer’s instructions.

### 2.4. Statistical Analysis

Continuous data are given as a median and interquartile range, and categorical data as counts and percentages. Nonparametric ANOVA (Kruskal Wallis test with Dunn’s multiple comparisons post-test) was used for comparison among vaccination regimens. Logistic regression, fixing a cut-off of IgG response at 2080 BAU/mL, was used in order to explore possible predicting factors of the IgG anti-S response. Statistical significance was assumed if *p* values were below 0.05. All the analyses were performed using SPSS 23.0 software (SPSS Inc., Chicago, IL, USA). GraphPad Prism (version 8.4.2, GraphPad Software, San Diego, CA, USA) was used to draw the box and whisker plots.

## 3. Results

### 3.1. SARS-CoV-2 TrimericS IgG Antibody Levels Two Months after Vaccination

All subjects developed a positive SARS-CoV-2 TrimericS IgG antibody response, with the following exceptions: 3 individuals out of 184 (1.6%) and 3 individuals out of 985 (0.3%) belonging to group ChAd/ChAd and BNT/BNT, respectively, presented antibody titer below 33.8 BAU/mL. A total of 336/1248 (27%) participants had a serological test result above the upper limit of quantification (>2080 BAU/mL) with the following distribution: 5/184 (3%) in group ChAd/ChAd, 289/985 (29%) in group BNT/BNT and 42/79 (53%) in group ChAd/BNT. The comparison of antibody titers among groups showed that the heterologous vaccination (group ChAd/BNT) induced a significantly higher humoral response (IgG-anti-S median [IQR] 2080 [1240–2080] BAU/mL), followed by the homologous mRNA vaccine schedule (group BNT/BNT) (1480 [923–2080] BAU/mL) and by the homologous adenovirus-based vaccine (group ChAd/ChAd) (267 [127–561] BAU/mL) (Kruskal Wallis test with Dunn’s multiple comparisons post-test, *p* < 0.0001), (Figure 1).

### 3.2. SARS-CoV-2 Antigen and Anti-N Antibodies Testing Results

To assess for asymptomatic SARS-CoV-2 infection, both healthcare workers and university staff were periodically monitored with rapid antigen test and anti-nucleocapsid (N) antibodies test, respectively. Since anti-N antibodies can only be detected after a natural infection, any positive result may indicate that a vaccinated subject was in contact with the virus. Three subjects (3/952, 0.3%) tested positive for the antigen test (confirmed by SARS-CoV-2 RNA PCR), all in the 1–2 months before the vaccination (November–December 2020). Anti-N IgM and IgG antibodies were discovered in 11 (11/296, 3.7%) and 6 (6/296, 2%) subjects, respectively. All subjects positive for IgM or IgG anti-N, resulted negative for SARS-CoV-2 RNA PCR. Three of the anti-N IgG positive subjects had declared a prior COVID-19 diagnosis.

### 3.3. Factors Affecting IgG Response

From those who had completely filled out a questionnaire within the Informed Consent Form [8] out of the 284 individuals (*n* = 175, 62% ChAd/ChAd; *n* = 32, 11% BNT/BNT; *n* = 77, 27% ChAd/BNT) we were able to collect a series of supplementary information in addition to the vaccine schedule on age, sex, BMI, smoking, diabetes, cardiovascular diseases, respiratory tract diseases, COVID-19 diagnosis, vaccine side effects, allowing a logistic regression analysis. The results showed that only the vaccine schedule significantly affected the antibody titers (*p* < 0.0001), (Table 2). Regarding the safety of COVID-19 vaccines, the majority of the subjects (170/284, 60%) declared no side effects; the remaining 114 subjects (40%) reported side effects that went away within 4 days.

## 4. Discussion

Safety considerations associated with the ChAdOx1-S vaccine have led some European countries (e.g., Italy) to recommend the switch from the homologous booster to a heterologous booster, such as BNT162b2. Several studies have been assessing the safety and efficacy of various combinations of heterologous prime-boost vaccination in clinical trials [4,9], and heterologous prime-boost strategies have been indicated to be immunogenic and safe [10,11]. The present study analyzed the antibody levels of three real-world cohorts of vaccinated participants with ChAd/BNT compared to homologous regimens (BNT/BNT and ChAd/ChAd).

We found that two months after vaccination, IgG levels of the heterologous ChAd/BNT group were significantly higher than those of the homologous groups (BNT/BNT and ChAd/ChAd) and that those of the BNT/BNT group were significantly higher than ChAd/ChAd, according also to recent reports [12,13]. The median antibody levels were only 8 times higher than the test-positivity cut-off in the ChAd/ChAd group, 44 times in the BNT/BNT group and up to 62 times in the ChAd/BNT group. Furthermore, after two months, at least one out of two vaccinated with ChAd/BNT reached an anti-S titer over 2080 BAU/mL (upper limit of assay). 

Moreover, we analyzed which factors could influence the IgG anti-S levels among vaccine schedule, sex, age, BMI, smoking, diabetes, cardiovascular diseases, respiratory tract diseases, COVID-19 diagnosis and vaccine side effect, in order to identify the role of each factor net of the others. Our results demonstrated that only the vaccine schedule significantly impacted antibody response and on the contrary to previously published papers [14,15], the age did not affect the humoral response. This may be due to the relatively young age of the cohort (78% of subjects are under 60 years old) and the small size of the subset (284 individuals) analyzed in the regression analysis. Regarding the safety of COVID-19 vaccines, the majority of the subjects declared no side effects; the remaining subjects have reported side effects that went away within a few days. It is noteworthy that no severe side effects were reported for either the homologous groups or the heterologous group.

This report is subjected to some limitations. Firstly, approximately 25% of our subjects had antibody titer, which was approximated to 2080 BAU/mL (upper limit of assay); however, since the censored data mainly concerned the heterologous schedule of vaccination, the proposed statistical approach seems also more conservative. Secondly, the impact of asymptomatic subjects on anti-S IgG levels, is not fully addressed, because we received only partial information about SARS-CoV-2 infection in the period between vaccination and antibody titer analysis, or also prior to vaccination. However, following the periodic testing of nucleocapsid-specific IgG antibodies in the university staff, and of SARS-CoV-2 antigens in the healthcare workers (followed by real-time PCR for SARS-CoV-2 RNA detection), to check for the possible contact with the virus over the course of the study, we minimized the effects of any asymptomatic infection on our results. Finally, our study did not account for other mechanisms of immune protection such as T-cell responses and their role in vaccine protection against severe infections of SARS-CoV-2 [16], nor did we perform a serum neutralization assay. However, the assay used for monitoring the anti-S-IgG titers shows a high concordance with the neutralizing IgG antibodies [6,7], confirming the effectiveness of the heterologous vaccination in stimulating the humoral immunity against COVID-19 [17]. In fact, it has been reported that the heterologous approach to the SARS-CoV-2 vaccination results in a complementary robust humoral and cellular immunity, which in turn will help to reduce the transmission of emerging variants and protect immunocompromised patients, including those with malignancies, transplants or frail people [17,18].

Differently from recent randomized and observational clinical studies [9,12,13], our investigation provides real-life data on the effectiveness of heterologous vaccination in a cohort of healthy workers, predominantly young to middle adulthood, belonging to the medical and academic setting in the short post-vaccination period. Taken together, these results showed that the heterologous vaccination is an effective and safe alternative to homologous vaccination to increase humoral immunity against COVID-19, providing further information for vaccination and logistical strategies. Moreover, the flexibility of schedules should be considered to improve access to COVID-19 vaccination globally.

These data assess the short-term humoral response (two months) induced by the vaccination over a 12-month course following the administration of the primary COVID-19 vaccination. The study will end in June 2022, and the long-term dynamic of SARS-CoV-2 anti-S-IgG titers of the heterologous group compared to homologous groups of vaccinated subjects may be of utmost importance to provide prospective real-life data about immunogenicity.

## Figures and Tables

**Figure 1 vaccines-10-00491-f001:**
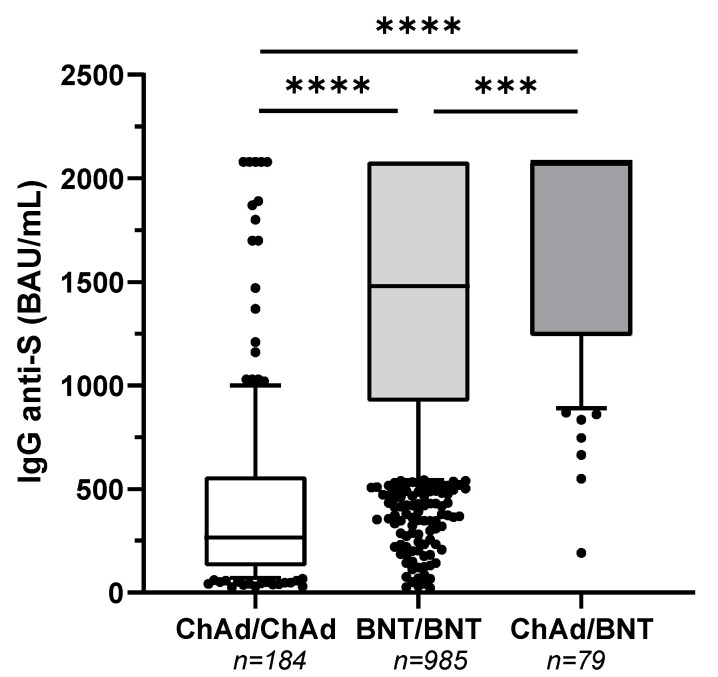
Heterologous ChAdOx1-S nCoV-19/BNT162b2 prime-boost vaccination enhances antibody titers. Comparison of anti-trimeric spike protein IgG antibodies to SARS-CoV-2 among three different groups of vaccinated subjects. ChAd/ChAd denotes an Oxford-AstraZeneca ChAdOx1-S nCoV-19 vaccine for prime and second doses. BNT/BNT denotes a Pfizer-BioNTech BNT162b2 vaccine for prime and second doses. ChAd/BNT denotes a ChAdOx1-S nCoV-19 vaccine for the prime dose and a BNT162b2 vaccine for the second dose. The serum samples were analyzed approximately two months after the second dose. Box-whisker plot displaying the 90/10 percentile at the whiskers, the 75/25 percentiles at the boxes, and the median in the center line. **** *p* < 0.0001, *** *p* < 0.001, Kruskal Wallis test with Dunn´s multiple comparisons post-test.

**Table 1 vaccines-10-00491-t001:** Characteristics of the three groups of vaccinated subjects.

Characteristic		All	ChAd/ChAd	BNT/BNT	ChAd/BNT
		*n* = 1248	*n* = 184 (15%)	*n* = 985 (79%)	*n* = 79 (6%)
Gender	Male	412 (33%)	83 (45%)	292 (30%)	37 (47%)
	Female	836 (67%)	101 (55%)	693 (70%)	42 (53%)
Age	Years (median, IQR)	52 (44–59)	55 (48–61)	51 (43–58)	52 (46–57)
Interval between prime and boost dose	Days (median, IQR)		74 (74–74)	25 (21–35)	83 (83–83)
Interval between boost dose and antibody titer test	Days (median, IQR)		61 (57–65)	60 (60–60)	61 (55–67)

ChAd/ChAd denotes a ChAdOx1 nCoV-19 (ChAd) COVID-19 vaccine (Vaxzevria, AstraZeneca) for prime and boost doses. BNT/BNT denotes BNT162b2 (BNT) COVID-19 vaccine (Comirnaty, Pfizer–BioNTech) for prime and boost doses. ChAd/BNT denotes a ChAd vaccine for prime dose and a BNT vaccine for boost dose.

**Table 2 vaccines-10-00491-t002:** Characteristics of a subgroup of 284 subjects included in the logistic regression analysis.

Factor	*n* = 175 (62%)	*n* = 32 (11%)	*n* = 77 (27%)	^3^*p* Value
Vaccine schedule	ChAd/ChAd;	BNT/BNT	ChAd/BNT	<0.0001
Sex (Male)	81 (46%)	11 (34%)	37 (48%)	0.965
^1^ Age	55 (48–61)	52 (43–60)	52 (46–57)	0.167
^1^ BMI	23.84 (21.8–27.08)	24.15 (21.65–27.15)	23.84 (21.99–26.12)	0.631
Smoking	18 (10%)	1 (3%)	15 (19%)	0.148
Diabetes	3 (2%)	5 (16%)	1 (1%)	0.411
Cardiovascular diseases	7 (4%)	3 (9%)	2 (3%)	0.127
Respiratory tract diseases	2 (1%)	3 (9%)	0 (0%)	0.747
COVID-19 diagnosis	4 (2%)	0 (0%)	4 (5%)	0.264
^2^ Vaccine side effects	66 (38%)	12 (38%)	36 (47%)	0.697

^1^ Median (IQR). ^2^ i.e., local pain, headache, malaise, fatigue, fever, feverish, muscle ache, joint pain, nausea as self-reported in the 48 h after vaccination. ^3^ Logistic regression analysis, fixing a cut-off of IgG response at 2080 BAU/mL (upper limit of assay).

## Data Availability

Not applicable.

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
