# Peer review of "Evaluation of Two-Month Antibody Levels after Heterologous ChAdOx1-S/BNT162b2 Vaccination Compared to Homologous ChAdOx1-S or BNT162b2 Vaccination"

_vaccines, 2022, doi:10.3390/vaccines10040491_

Round 1

Reviewer 1 Report

In the current manuscript, the authors analyzed the effects of vaccination using heterologous mRNA vaccines on the titers of IgG for anti-spike (IgG anti-S) protein of COVID-19.  The manuscript is well summarized, and the result is important to understand the effects of vaccination.  One thing that the authors need to consider is that there are many cases that showed low titers of IgG anti-S in the group of homologous BNT/BNT (Figure 1).  It is therefore unclear whether the values in BNT/BNT followed normal distribution or not. Since this point influences the statistical analysis, it needs to evaluate the sample distribution of the group.

Reviewer 2 Report

The brief communication by Barocci, Orlandi and colleagues is the report of a population survey on Sars-CoV-2 antibodies produced after vaccination.
The introduction is very brief and not very informative, but this is acceptable considering the nature of the publication.
The methods section os also very brief, and in particular doesn't provide any information about the experimental assay protocol, nor about the statistical
analysis which is later presented in the results. It is correct but not sufficient to cite the used sotware, there should be indication of the used methods. Please expand the section adding relevant information.
The results are based on a single assay, and this is a strong limit. Adding a validation test on a subset of samples would have much enriched the value of the study (i.e.: serum neutralization assay). 
Also, it would be precious adding information about the in vivo efficacy of the vaccination, by gathering and showing some follow up epidemiological data (how the different vaccinated groups were effectively protected in the following months?).
I mean, two indications should be provided:1) Were the subjects infected with SARS-CoV-2 between the two doses, or between the second dose and the test? (this would probably affect the results); 2) were the subjects infected after the study? In which proportion
among the three groups?
The entire section 3.2 is based on a statistic analysis of data coming from a subset of cases, but there is no indication of how this subset was selected. There should be some information about this. Table 2 reports the results of the variance analysis
but should also show the abolute values of the indicated parameters divided by groups, to provide the reader with relevant information.
Figure 2 reports the same analysis of figure 1, limited to the subset. It is not inoformative, in my view, since we don't have indications of the subsampling criteria, but shows a bias against the bnt/bnt group, which is neatly underrepresented
Last, the discussion is appropriate. At the end, there is indication of a forecoming publication with follow up data (including, I guess, those which I was recommending to add here). Please consider that in my opinion data presented in this publication
would not be usable again for this expected new one, as for the current scientific publication guidelines. 
A few minor points:
line 18: the abstract reports the abbreviated name of vaccines, while it would be better to use extended or commercial names
line 52: 'in the northern area of Marche Region'
line 56-59: this sentence is not appropriated for this section, it should be considered for the discussion.
line 65: '(PU)'
line 67: please omit or rephrase 'based on a technical datasheet'
table 1: please uniformate choosing between (median, IQR) and (mean ± SD).
line 84: please add information about manufacturer company
line 84: the reported data about sensitivity and specificity don't come from the study, but from literature. Please specify this more clearly in the text and cite an appropriate reference.
line 91: the reported data come from literature. Please specify this more clearly in the text.The cited reference is a manufacurer's brochure, I recommend a more appropriated literature citation
line 101:the cut off of 33.8 BAU/mL is not described in the methods section
line 134: figure 2 shows data which are not detailed in the main text. Please add the data details as done for figure 1.
line 161: please avoid this consideration: the underestimation could be avoided by further diluting the samples, which you didn't perform.
line 174: these considerations about the methods used should be supported by something else than the manufacturers brochure. Please adda a different supporting reference and/or rephrase.
line 187: please check for authors contributions: the mere 'funding acquisition' is not a sufficient criterium for authorship. This should instead considered in the funding and/or acknowledgments sections.
line 189: 'SPSAL' is possibly 'SPISAL'?
line 203: please check for reference style
line 215: please check for reference style

Reviewer 3 Report

  1. The reviewer does not see any novelty of this study. Please comment on this in the discussion.
  2. The analytical kit used for the determination of the antibody titer has an upper limit of 2080 BAU/ml. Approximately 25% of all patients had the result above this limit and their titer was artificially set at 2080. Thus, quantitative comparisons of the antibody titer between groups are subject to critical error. A possible remedy for this error is to retest the antibody titers in these samples applying  dilutions. Alternatively, you can exclude these samples from the analysis or perform a dichotomous analysis of the antibody titer in groups above / below a certain limit (high vs. low titer).
  3. It cannot be ruled out that the level of anti-s antibodies is some extent induced by SARS-CoV-2 infection (e.g. asymptomatic) which could have occurred in the period between vaccination and analysis (after 2 months). The infection could also have occurred prior to vaccination and the humoral response could have come from existing memory cells. This could be another  fatal study error. How the authors ruled out this possibility?
  4. How were side effects analyzed after vaccination? What scale was used? The FDA Center for Biologics Evaluation and Research scale used in most pivotal studies on vaccine safety may be referenced (eg. Polewska et all: Medicina 202157(7), 732).

  5. Really age had no significant influence on the humoral response? This is a surprising finding, possibly due to the fact that the titer analysis was artificially limited to 2080 BAU / ml.
  6. Why was the antibody determination performed after 2 months ? Two months after vaccination, the disappearance of antibodies can already be observed.
  7. What might be the clinical relevance of the study results? For which patients can heterologous vaccination be useful? 

Round 2

Reviewer 2 Report

The paper has been substantially modified, according to most of my considerations. In my opinion it is now improved and can be accepted for publication

Reviewer 3 Report

The authors' responses and the changes made to the manuscript are satisfactory for the reviewer.